# SPHINX: Structural Prediction using Hypergraph Inference Network

**Iulia Duta** [1]    **Pietro Liò** [1]

## Abstract

The importance of higher-order relations is widely recognized in numerous real-world systems. However, annotating them is a tedious and sometimes even impossible task. Consequently, current approaches for data modelling either ignore the higher-order interactions altogether or simplify them into pairwise connections. To facilitate higher-order processing, even when a hypergraph structure is not available, we introduce SPHINX, a model that learns to infer a latent hypergraph structure in an unsupervised way, solely from the final task-dependent signal. To ensure broad applicability, we design the model to be end-to-end differentiable, capable of generating a discrete hypergraph structure compatible with any modern hypergraph networks, and easily optimizable without requiring additional regularization losses. Through extensive ablation studies and experiments conducted on four challenging datasets, we demonstrate that our model is capable of inferring suitable latent hypergraphs in both transductive and inductive tasks. Moreover, the inferred latent hypergraphs are interpretable and contribute to enhancing the final performance, outperforming existing methods for hypergraph prediction.

## 1. Introduction

Graphs are universally recognized as the standard representation for relational data. However, their capabilities are restricted to modelling pairwise connections. Emerging research shows that real-world applications including neuroscience (Guo et al., 2021), chemistry (Jost & Mulas, 2018), biology (Viñas et al., 2022), often exhibit group interactions, involving more than two elements. This leads to the development of a new field dedicated to representing higher-order relations, in the form of hypergraphs. However, while graph datasets are widespread in the machine learning community (Leskovec et al., 2005; Hu et al., 2020), the availability of hypergraph datasets is much more limited. Recent work (Wang & Kleinberg, 2024) highlights two potential causes for the lack of higher-order data. On one hand, current technology used for collecting information is mostly designed or optimised to detect pairwise interactions. Furthermore, even in the exceptional case when the data is gathered in a higher-order format, the published version is often released in a reduced, pairwise form.

Therefore, to preserve the higher-order information, it is crucial to develop methods for learning the hypergraph structure in an unsupervised way, only from point-wise observations. To create a general, widely-applicable model, we identify a set of key desiderata. 1) **The model should be applicable to a broad type of tasks.** This is a challenge for most existing hypergraph predictors, which optimize a single hypergraph, limiting their usefulness to transductive setups. 2) **The model should be compatible with any hypergraph processing architecture.** This requires our inference model to be fully differentiable while producing a sparse and discrete hypergraph structure. Existing methods often fall short, as they are either only partially differentiable (relying on techniques such as top-k selection or thresholding to enforce sparsity) or restricted to generating only weighted hypergraphs. 3) **A powerful model should be easy to optimise.** This is a limitation exhibited by most of the modern architectures for which predicting a suitable hypergraph requires various types of regularization losses.

Motivated by these, we introduce Structural Prediction using Hypergraph Inference Network (**SPHINX**), a model for unsupervised latent hypergraph inference, that can be used in conjunction with any recent models designed for hypergraph processing. SPHINX models the hyperedge discovery as a clustering problem, adapting a soft clustering algorithm to *sequentially* identify subsets of highly correlated nodes, corresponding to each hyperedge. To produce a discrete hypergraph structure, we take advantage of the recent development in *differentiable k-subset sampling* (Ahmed et al., 2023; Minervini et al., 2023), obtaining a more effective training procedure, that eliminates the necessity for heavy regularisation. While classical selection methods such as Gumbel-Softmax (Jang et al., 2017) fail to control the sparsity of the hypergraph, our constrained k-subset sampling produces a more accurate latent structure.

---

[1]Department of Computer Science, University of Cambridge. Correspondence to: Iulia Duta <id366@cam.ac.uk>.

*Proceedings of the 42ⁿᵈ International Conference on Machine Learning*, Vancouver, Canada. PMLR 267, 2025. Copyright 2025 by the author(s).

Experiments on both inductive and transductive datasets show that our inferred hypergraph surpasses existing graph and hypergraph-based models on the final downstream task. Moreover, our synthetic experiments reveal that the inferred structure correlates well with the ground-truth connectivity that guides the dynamical process, outperforming other hypergraph predictors. The resulting model is general, easy to optimise, which makes it an excellent candidate for modelling higher-order relations, even in the absence of an annotated connectivity.

**Our main contributions** are summarised as follows:

1. We propose a **novel method for explicit hypergraph inference**, that uses a sequential predictor to identify subsets of highly-related nodes and a $k$-subset sampling to produce an explicit hypergraph structure, that can be plugged into any hypergraph neural network.

2. The model performs **unsupervised hypergraph discovery**, by using supervision only from the weak node-level signal. We empirically show that the predicted hypergraph correlates well with the true higher-order structure, even if the model was not optimised for it.

3. The latent hypergraph enforces an **inductive bias for capturing higher-order correlations**, even in the absence of the real structure, which proved to be **beneficial for downstream tasks** such as trajectory prediction (inductive) or node classification (transductive). Having an explicit structure allows us to visualise the discovered hypergraph, **adding a new layer of interpretability to the model**.

## 2. Related work

**Structural inference on graphs.** Modelling relational data using Graph Neural Networks (GNNs) (Kipf & Welling, 2017; Veličković et al., 2018) proves to be beneficial in several real-world domains (Bica & van der Schaar, 2022; Sanchez-Gonzalez et al., 2020; Lam et al., 2023). Most of the current graph methods assume that the graph connectivity is known. However, providing the relational structure is a highly challenging task, that, when possible to compute, requires either expensive tools, or advanced domain knowledge. These limitations led to the development of a new machine learning field dedicated to learning to infer structure from data in an unsupervised way. Neural Relational Inference (Fetaya et al., 2018) is one of the pioneering works inferring an adjacency matrix from point-wise observations. The model consists of an encoder that learns to predict a distribution over the potential relationships and a GNN decoder that receives a sampled graph structure and learns to predict the future trajectory. fNRI (Webb et al., 2019) extends this work by inferring a factorised latent space, capable of encoding multiple relational types, while (Löwe

et al., 2022) incorporates causal relations into the framework. For temporal data, these models infer a single structure for the entire timeseries. dNRI (Graber & Schwing, 2020) improves on that respect, by modifying the graph structure at each timestep.

**Hypergraph Networks.** Several data structures for higher-order modelling were proposed, including simplicial complexes (Torres & Bianconi, 2020), cell complexes (Lundell & Weingram, 1969), and more generally, hypergraphs (Banerjee, 2021). To model hypergraphs, several deep learning methods have recently emerged. Hypergraph Neural Networks (Feng et al., 2019) applies GNNs on top of a weighted clique expansion. This could be seen as a two-stage message passing scheme, sending messages from nodes to hyperedges and viceversa. UniGNN (Huang & Yang, 2021) and AllDeepSets (Chien et al., 2022) introduce a more general framework in which the two stages can be implemented as any permutation-invariant function such as DeepSets (Zaheer et al., 2017) or Transformers (Vaswani et al., 2017). Further proposed extensions include the integration of attention mechanisms (Bai et al., 2021; Zhang et al., 2022; Georgiev et al., 2022), or attaching additional geometric structure to the hypergraph, through the incorporation of cellular sheaves (Duta et al., 2023). While these methods can be seen as learning to modify the structure, they require an initial hypergraph structure.

**Structural inference on hypergraphs.** Providing the hypergraph structure to the model requires measuring higher-order correlations, which often implies a highly difficult and expensive annotation process. This leads to the necessity of inferring the latent hypergraph structure directly from data. However, when moving from the graph realm to the hypergraph domain, the set of potential edge candidates abruptly increases from quadratic to exponential. This makes the problem of inferring the latent hypergraph structure significantly more challenging. Classical attempts of achieving this include methods based on Bayesian inference (Young et al., 2021) or statistical approaches that filter against a null model for hypergraphs (Musciotto et al., 2021).

While relational inference for graphs is an established field in deep learning, it remains largely underexplored in the hypergraph domain. The common approach is to either extract hyperedges using k-NN (Huang et al., 2009), clustering (Jiang et al., 2019; Gao et al., 2013) or iteratively optimising regularisation constraints (Liu et al., 2016). However, the resulting hypergraph is static, independent of the targeted downstream task. More recent several methods (Yu et al., 2012; Wang et al., 2024; Cai et al., 2022; Zhang et al., 2018; Gao et al., 2020) improve on the kNN approach by learning a task-guided mask to modify the initial structure. While producing dynamic hypergraphs, optimising the mask restrict the method to only work in the transductive setup, in

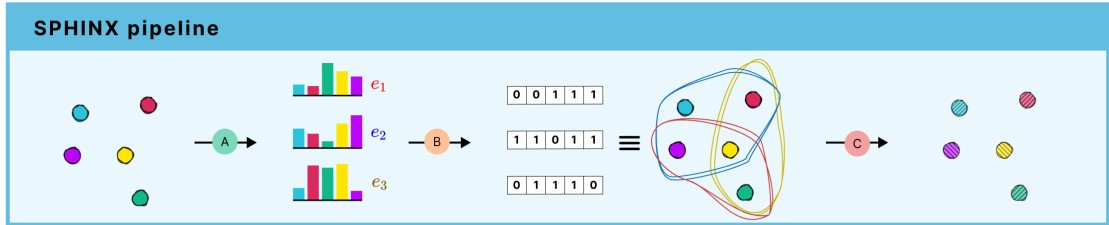

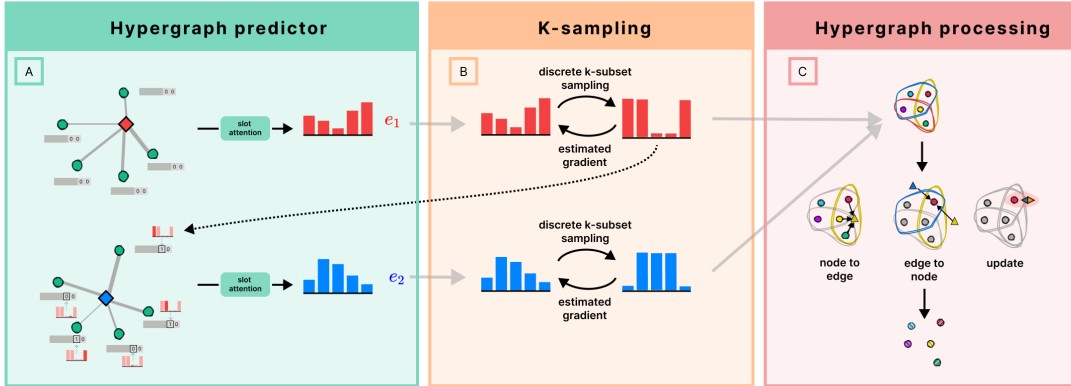

Figure 1: **SPHINX architecture**, designed to capture higher-order interactions without having access to the ground-truth hypergraph structure. The model consists of three stages: A) The **hypergraph predictor** infers a probabilistic latent hypergraph structure, by using a sequential clustering algorithm to produce global, history-aware hyperedges. At each timestep, the node features are enriched with the information about the already predicted hyperedges and the resulting cluster represent the hyperedge membership. B) A $k$-**subset sampling** algorithm is applied to transform each probability distribution into discrete incidence relations, while maintaining the end-to-end differentiability of the framework. The $k$-nodes constraint ensures a more stable optimisation process, beneficial for the final performance. C) The predicted hypergraph structure can be used in any standard **hypergraph neural network** in order to produce higher-order representations.

which a single hypergraph is produced for the entire dataset.

Very few methods in the literature are general enough to allow learning hypergraph structure in both inductive and transductive tasks.[1] GroupNet (Xu et al., 2022) and DynGroupNet (Xu et al., 2024) use the correlation matrix to extract submatrices of high connectivity, EvolveHypergraph (Li et al., 2022) predicts a soft connectivity for each node, relying on Gumbel-Softmax and regularisation tricks to produce a sparse, discrete structure, while TDHNN (Zhou et al., 2023) uses a differentiable clustering to produce soft incidence, followed by a top-k selection to impose sparsity. For a more in-depth analysis of the existing dynamic hypergraph predictors, please see Section B of the Appendix.

## 3. Structure Prediction using SPHINX

The Structure Prediction using Hypergraph Inference Network (SPHINX) model is designed to produce higher-order representations without access to a ground-truth hypergraph structure. From the set of point-level observations, the model learns to infer a latent hypergraph that can be further used in conjunction with any classical hypergraph neural network architecture.

Our input consists of a set of nodes $V = \{v_i | i \in \{1 \ldots N\}\}$, each one characterised by a feature vector $x_i \in \mathbb{R}^f$. The goal is to predict, for each node, the target $y_i \in \mathbb{R}^c$. In the entire paper, we are following the assumption that the node-level target $y_i$ is the result of a higher-order dynamics, guided by an unknown higher-order structure $\tilde{\mathcal{H}}$.

The processing stages of our method are summarized in Figure 1. First, we predict the latent hypergraph based on the input features $\mathbf{X}$. The inferred hypergraph structure is then fed into a hypergraph network to obtain the final prediction. To fulfil our goal of obtaining a generally applicable method, easy to optimize and compatible with any hypergraph architecture, our method needs to satisfy two important characteristics. Firstly, the pipeline needs to be end-to-end differentiable, such that the latent hypergraph inference can be trained in a weakly supervised fashion, solely from the downstream signal. Secondly, it needs to produce a sparse and discrete structure, such that it can be used inside any existing hypergraph processing model.

---

[1]From now on we use the term *inductive* when a different hypergraph is predicted for each example and *transductive setup* when a single hypergraph is predicted for the entire dataset.

To achieve these, we designed the model using two core components: a learnable soft clustering that *sequentially* predicts a probability distributions for each one of the $M$ potential hyperedges, and *a differentiable $k$-sampling* that, based on this probability distribution, samples discrete subsets of $k$ nodes forming the hypergraph structure. Both components are differentiable such that the model can be easily trained using standard backpropagation techniques. Moreover, the predicted hypergraph structure is a discrete object, following the classical structural representation used in any hypergraph processing models $\mathcal{H} = (V, E)$, where $V$ is a the of nodes and $E$ is a the of hyperedges.

### 3.1. Hypergraph predictor

The goal of this module is to transform a set of node features $\mathbf{X} \in \mathbb{R}^{N \times f}$ into a set of incidence probabilities $\mathbf{P} \in \mathbb{R}^{N \times M}$ where $N$ represents the number of nodes and $M$ represents the expected number of hyperedges. Each column $j \in \{1 \dots M\}$ corresponds to a hyperedge and represents the probability of each node being part of the hyperedge $j$. In order to accurately predict these probabilities, the model needs to have a global understanding of the nodes interactions and identify the subsets that are more likely to exhibit a higher-order relationship.

Taking inspiration from the computer vision literature for unsupervised object detection (Greff et al., 2020), we model the hypergraph discovery task as a soft clustering problem, where each cluster corresponds to a hyperedge. We adapt the iterative slot-attention algorithm (Locatello et al., 2020) to produce $M$ clusters, each one corresponding to a predicted hyperedge. The probability of a node $i$ being part of a cluster $j$, computed as the node-cluster similarity, represents the incidence probabilities $p_{ij}$ corresponding to node $i$ and hyperedge $j$. Therefore, we will use the terms slots and hyperedges interchangeably.

**Slot Attention for probabilistic incidence prediction.** We start by creating $M$ slots, one for each hyperedge. Each slot $s_j \in \mathbb{R}^f$ is randomly initialized from a normal distribution. At each iteration, the slots representation is updated as the weighted average of all the nodes, with the weights computed as a learnable dot-product similarity as indicated by Equation 1, where $f_1$, $f_2$ and $f_3$ represent MLPs and $\sigma$ is a non-linearity. After $Q$ iterations, the pairwise similarity between the updated slot representation and the set of node features represents the predicted probability distribution $p_{:,j}$ for hyperedge $j$.

$$s_j^{q+1} = \sum_i \left( \sigma(f_1(s_j^q)^T f_2(x_i)) f_3(x_i) \right) \quad (1)$$

$$p_{i,j} = p(v_i \in e_j) = \sigma(f_1(s_j^Q)^T f_2(x_i)) \quad (2)$$

**Sequential Slot Attention for solving ambiguities.** The above algorithm suffers from a strong limitation. Due to the symmetries exhibited by the set of hyperedges, independently inferring $M$ hyperedges leads to strong ambiguity issues. Concretely, since the slots are initialized randomly, there is no mechanism in place to distribute the hyperedges between slots: multiple slots could be attached to the same obvious hyperedge, leaving others completely uncovered.

To alleviate this, we propose a sequential slot attention. Instead of predicting all hyperedges simultaneously, we will predict them sequentially, one at a time, ensuring that at each timestep the hyperedge prediction mechanism is aware of the hyperedges predicted so far. To achieve that, the features of each node $x_i$ are enriched with an additional binary vector $b_i \in \{0, 1\}^{(M-1)}$ indicating the relationship between that node and the previously predicted hypergraphs. Specifically, when predicting hyperedge $j$, for each previous hyperedge $t$ $(t < j)$, $b_{it} = 1$ if the node $i$ was previously selected to be part of the hyperedge $t$ and $b_{it} = 0$ otherwise. This way, the slot-attention algorithm has the capacity to produce more diverse hyperedges, as we show in Section 4.1.

### 3.2. Discrete constrained sampling

The hypergraph predictor module, as described above, produces a probabilistic incidence matrix $\mathbf{P} \in \mathbb{R}^{N \times M}$, where each element $p_{i,j}$ denotes the probability of a node $i$ being part of the hyperedge $j$. However, the standard hypergraph neural network architectures are designed to work with sparse and discrete rather than probabilistic structures.

Previous work (Li et al., 2022) employs Gumbel-Softmax (Jang et al., 2017) to differentiably sample from a categorical distribution. However, these techniques sample each element in the hyperedge independently, without any control on the cardinality of the hyperedge. This leads to unstable optimisation, that requires additional training strategies such as specific sparsity regularisation. Top-k is another popular approach to achieve sparsity (Zhou et al., 2023; Xu et al., 2022). However, to preserve gradients, the generated structure cannot be discretized. Although a weighted hypergraph can be beneficial for certain architectures, it is not universally compatible with all decoders and makes the interpretation of the latent space more challenging.

To address this issue, we are leveraging the recent advancement in constrained $k$-subset sampling (Ahmed et al., 2023; Niepert et al., 2021; Minervini et al., 2023). These methods were successfully used to tackle discrete problems such as combinatorial optimisation, learning explanations and, more recently, rewiring graph topology (Qian et al., 2023). Different from classical differentiable samplers (Jang et al., 2017), the $k$-subset sampler would produce a subset of size exactly $k$, equipped with a gradient estimator useful for backpropagation.

In our work, we took advantage of these recent advance-

ments and apply it to produce a discrete incidence matrix from the probabilities inferred by the slot attention algorithm. Concretely, given the probability distribution $p_{:,j}$ for each hyperedge $j$, the discrete sampler would select a subset of nodes, representing the group of nodes forming the hyperedge. As demonstrated in Section 4.1, by ensuring that each hyperedge contains exactly $k$ elements, this approach improves over the previous techniques, manifesting an easier optimisation. While the cardinality $k$ needs to be set apriori, as a hyperparameter, this value can vary between different hyperedges.

## 3.3. Hypergraph processing

Producing a discrete hypergraph structure and being able to propagate the gradient through the entire pipeline, enable us to process the resulting latent hypergraph with any existing architecture designed for higher-order representations. In recent years, several architectures were developed for hypergraph-structured input (Feng et al., 2019; Wang et al., 2022; Huang & Yang, 2021; Chien et al., 2022). Most of them follow the general two-stage message-passing framework. In the first stage, the information is sent from nodes to the hyperedges using a permutation-invariant operator $z_j = f_{V \to E}(\{x_i | v_i \in e_j\})$. In the second stage the messages are sent back from hyperedge to nodes $x_i = f_{E \to V}(\{z_j | v_i \in e_j\})$.

In our experiments we use a similar setup to the one proposed in (Chien et al., 2022), in which the two functions $f_{V \to E}$ and $f_{E \to V}$ are implemented as DeepSets (Zaheer et al., 2017) models $f_{V \to E}(S) = f_{E \to V}(S) = \text{MLP}(\sum_{s \in S}(\text{MLP}(s)))$. Note that, although we experimented primarily with the architecture presented above, the framework is general enough to be used with any other hypergraph network. Our experiments demonstrated that even simpler hypergraph models enable the discovery of accurate higher-order structures.

## 4. Experimental Analysis

While the importance of higher-order processing is widely accepted in the machine learning community (Zhou et al., 2006) and beyond (Zhang et al., 2020), the amount of benchmarks developed to properly validate the hypergraph methods is still insufficient. This issue becomes even more pronounced when it comes to latent hypergraph inference. Although there is evidence that many existing real-world tasks exhibit underlining higher-order interactions that can be beneficial to capture, evaluating the capability of a neural network to predict the latent structure remains challenging. First, the input features should contain enough information to predict the hypergraph structure. Secondly, even if we do not need the hypergraph structure as a supervision signal, having access to it is necessary to properly evaluate to what extent the model learns to infer the correct structure.

To alleviate these shortcomings, current works adopt one of the following approaches: either using synthetic data, where we can directly have access to the higher-order relationships used to generate the outcome, or by using real-world data and only evaluate the capability of the model to improve the final prediction, without directly computing the accuracy of the discovered latent structure.

In our work we adopt both approaches. First, we perform an in-depth ablation study on a synthetic dataset containing particle simulations. Then, we move to the real-world datasets where we evaluate on both inductive and transductive setups. Our goal is two-fold. We show that our model is capable of learning an appropriate latent hypergraph structure in an unsupervised way. Additionally, we prove that the latent higher-order structure, learnt jointly with the rest of the model, helps the performance of the downstream task on both the inductive and transductive problems.

### 4.1. Particle Simulations

Our simulated system consists of $N$ particles moving in a 2D space. For each example, $K$ random triangles were uniformly sampled to represent $K$ 3-order interactions. All the particles that are part of a triangle rotate around the triangle's center of mass with a random angular velocity $\theta_i$ characteristic to each particle $i$. If one particle is part of multiple triangles, the rotation will happen around its average center of mass. Examples of such trajectories are depicted in Figure 3 and Figure 6. For each trajectory we observe the position of the particles in the first 22 steps, and the task is to infer the trajectory for the following 25 steps. Each example exhibits two higher-order interactions. The dataset contains 1000 trajectories for training, 1000 for validation and 1000 for test.

For all experiments, the *hypergraph predictor* treats particles as nodes. For each particle, the node features consist of the $(x, y)$ coordinates corresponding to the first $T$ timesteps. These features will be used to predict the incidence matrix $H$. During training, the *hypergraph processor* receives the predicted hypergraph structure together with the particle's position at timestep $t$, and the goal is to predict the position of the particles at timestep $t + 1$. To generate longer trajectories, during testing, the model receives at each moment in time the position predicted at the previous timestep.

**Hypergraph discovery.** These datasets allow us to experiment with learning the higher-order relations in a setup where we have access to the ground-truth connectivity. Our model produces discrete hypergraph structures that we can inspect and evaluate, offering an additional level of interpretability to the framework. Visualisations of our learned hypergraph structure (see Figure 3) reveal that they are

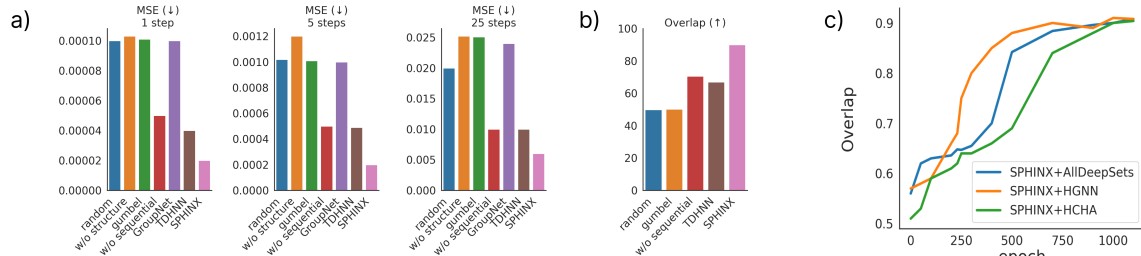

Figure 2: **Ablation studying the importance of hypergraph inference on the Particle Simulation. (a)** The sequential prediction and constrained $k$-subset sampling clearly helps the downstream performance. **(b)** SPHINX also predicts more accurate hypergraphs in terms of overlap. **(c)** Moreover, hypergraph discovery improves during training on the synthetic datasets, even if the model is not supervised for this task. Regardless of the hypergraph architectures, the models achieve a high overlap between the (unsupervised) predicted hypergraph and the gt. connectivitys.

highly correlated to the ground-truth interactions used to generate the dataset.

We quantitatively evaluate to what extent SPHINX learns the true higher-order relationships, by computing the overlap between our model's prediction and the ground-truth hyperedges. We refer to the Appendix for a full description of the metric used in this experiment. In Figure 2.c we observe that, even if we do not explicitly optimise for this task, the accuracy of the hypergraph structure predictor increases during training, reaching up to 90% overlap.

**General and versatile**, our model is designed to predict hypergraph structure that is useful for any hypergraph neural network. However, since the supervision signal comes from the high-level tasks, the architecture used for hypergraph processing can impose certain inductive biases, influencing the hypergraph structure learned by the predictor. In the experiment depicted in Figure 2.c, we investigate to what extent our model learns an accurate higher-order structure, regardless of the hypergraph processing architecture. We train a set of models using the same hypergraph predictor for inferring the hypergraph structure, but various hypergraph networks for processing it. For this, we experiment with AllDeepSets (Chien et al., 2022), HGNN (Feng et al., 2019) and HCHA (Bai et al., 2021). The results show that our model is capable of learning the suitable structure irrespective of the specifics in the processing model.

**Importance of sequential prediction.** While the classical slot-attention algorithm infers all the clusters in parallel, we argue that this is not an appropriate design choice for hypergraph prediction. Often, the set of hyperedges is mostly symmetric, which makes the problem of attaching the slots to different hyperedges highly ambiguous. In fact, we visually observed that, when predicting all the hyperedges simultaneously, the model tends to associate multiple slots to the same hyperedge, leaving others completely uncovered (see *SPHINX w/o sequential* model in Figure 6). This behaviour leads to both a decrease in downstream performance and a less accurate hypergraph predictor. As described in

Section 3.1, SPHINX alleviates this issue by inferring the slots sequentially, and providing at each step historical information about the structure predicted so far. Figure 2.a and 2.b demonstrate the benefits of our design choices. Sequential prediction (*SPHINX*) has a clear advantage, outperforming the classical simultaneous predictor (*SPHINX w/o sequential*).

**Influence of $k$-sampling.** Previous work on inferring hypergraphs (Li et al., 2022) takes inspiration from the graph domain (Fetaya et al., 2018) where the differentiable sampling is achieved using the Gumbel-Softmax trick. We argue that, inferring the subset of nodes that are part of a hyperedge generates more challenges compares to the graph scenarios. At the beginning of training, sampling each node-hyperedge incidence independently, without constraints, can lead to hyperedges containing either too few or too many nodes, which highly damages the optimisation. We can observe this phenomenon visually in Figure 6, column *SPHINX w gumbel*. In this work, we leverage the recent advance in differentiable $k$-subset sampling, which imposes the $k$-nodes constraint, while still allowing us to estimate the gradient during backpropagation. The experiments in Figure 2.a and Figure 2.b (see *gumbel* model vs *SPHINX* model) show that the $k$-subset sampling improves the results, leading to an easier optimisation and better final performance. This is in line with the results from (Li et al., 2022) where smoothness and sparsity regularisation are crucial for improving the results on a real-world dataset. In contrast, our model obtains competitive performance without any optimisation trick.

**Comparison with inductive hypergraph predictors.** The Particle Simulation dataset heavily relies on higher-order processing. In order to predict the particle movement, the model needs to identify the higher-order structures that guide the rotation. To understand to what extent our model is able to produce and process a useful hypergraph structure, we compare against two baselines: a model using a random structure (denoted as *random* in Figure 2) and a node-level model, that ignores the hypergraph structure (denoted as *w/o*

Table 1: **Ablation study on the NBA dataset**. Both using sequential, history-aware predictors and using $k$-subset sampling prove to be beneficial for the final performance. Moreover, SPHINX shows a clear advantage compared to the existing inductive hypergraph predictors on the single-trajectory prediction task (ADE/FDE metrics).

|  | 1SEC | 2SEC | 3SEC | 4SEC |
|---|---|---|---|---|
| SPHINX W/O SEQUENTIAL | 0.62/0.94 | 1.18/2.08 | 1.73/3.14 | 2.25/4.08 |
| SPHINX W GUMBEL | 1.29/1.81 | 2.10/3.34 | 2.81/4.60 | 3.45/5.66 |
| TDHNN | 0.68/1.03 | 1.27/2.22 | 1.84/3.31 | 2.30/4.29 |
| GROUPNET | 0.65/1.03 | 1.38/2.61 | 2.15/4.11 | 2.83/5.15 |
| SPHINX | **0.59/0.92** | **1.12/2.06** | **1.65/3.13** | **2.14/4.09** |

Table 3: **Performance in the inductive setup.** in terms of minADE$_{20}$ and minFDE$_{20}$ metrics for the NBA dataset.

|  | METHOD | 1SEC | 2SEC | 3SEC | 4SEC |
|---|---|---|---|---|---|
| GRAPH | NRI | 0.51/0.74 | 0.96/1.65 | 1.42/2.50 | 1.86/3.26 |
|  | DNRI | 0.59/0.70 | 0.93/1.52 | 1.38/2.21 | 1.78/2.81 |
|  | EVOLVEGRAPH | 0.35/0.48 | 0.66/0.97 | 1.15/1.86 | 1.64/2.64 |
|  | STGAT | 0.45/0.66 | 0.87/1.41 | 1.28/2.08 | 1.69/2.66 |
|  | TRAJECTRON++ | 0.44/0.67 | 0.79/1.18 | 1.51/2.49 | 2.09/3.52 |
| HYPERG. | EVOLVEH (H) | 0.49/0.74 | 0.95/1.68 | 1.44/2.27 | 1.91/3.08 |
|  | EVOLVEH (H+G) | 0.33/0.49 | 0.63/0.95 | 0.93/1.36 | 1.21/1.74 |
|  | TDHNN | 0.68/1.03 | 1.27/2.22 | 1.84/3.31 | 2.30/4.29 |
|  | GROUPNET | 0.34/0.48 | 0.62/0.95 | **0.87/1.31** | **1.13/1.69** |
|  | SPHINX | **0.30/0.43** | **0.59/0.94** | 0.88/1.38 | 1.16/1.74 |

*structure*). Both models perform much worse than our learnable hypergraph model, indicating that the dataset highly depends on higher-order interactions and that our model is capable of discovering an appropriate latent structure.

We also compared against existing hypergraph predictors in the literature. Most of the existing methods are constrained to only work in the transductive setup (by only predicting a single hypergraph for all examples). The only models that are applicable in the inductive setup are GroupNet (Xu et al., 2022), TDHNN (Zhou et al., 2023) and EvolveHypergraph (Li et al., 2022)[2]. The results in Figure 2.a show that our predictor outperforms all the existing ones. We believe that the drop in performance for the existing models is due to the poor gradients used in updating the predictor. Both GroupNet and TDHNN models use a form of top-k selection which only partially propagates the gradients. Moreover, qualitative evaluation shows that TDHNN suffers from the same limitation as our "w/o sequential baseline": all the hyperedges ended up selecting the same set of nodes.

Our model not only outperforms the existing ones in terms of downstream performance, but also discovers more accurate hyperedges. Computing the overlap between the prediction and the oracle hyperedges, SPHINX achieves an overlap of 90% compared to 67% for TDHNN (Figure 2.b)[3].

**Implementation details.** Unless otherwise stated, experiments use two slots and the AllDeepSets model as a decoder. For estimating the gradients in the $k$-subset sampling algorithm we experiment with SIMPLE (Ahmed et al., 2023), AIMLE (Minervini et al., 2023) and IMLE (Niepert et al., 2021) algorithms. The first two perform on par, while IMLE suffers from gradient issues. We use Adam optimizer for 1000 epochs, trained on a single GPU.

**Key findings.** The results presented in this section demonstrate that: ① Our model predicts accurate hypergraph connectivity, that correlates well with the ground-truth structure, even without being optimized for this task and ② ir-

[2]EvolveHypergraph does not have public code.

[3]For GroupNet we can not compute the overlap since the number of edges is constrained to be equal to the number of nodes.

respective of the hypergraph architecture used as decoder. ③ Predicting the hyperedges sequentially ameliorate the ambiguity issue, improving the hypergraph inference. ④ $k$-subset sampling is crucial for a good performance, proving to be superior to the standard Gumbel-Softmax approach and allowing us to eliminate the regularisation tricks. ⑤ Finally, discovering the latent hypergraph structure is clearly beneficial for the downstream prediction, improving upon existing inductive hypergraph predictors.

### 4.2. Inductive setup

To evaluate our model on an inductive real-world dataset, where we need to predict a distinct hypergraph for each example, we use the NBA SportVU dataset, containing information about the movement of the players during basketball matches. Each example contains 11 trajectories, 5 players from each team and one trajectory for the ball. The hypergraph predictor receives as node features the first 5 timesteps of each trajectory and the goal is to predict the next 10 steps. While our model is not especially designed as a tracking system, the dynamics followed by the basketball game contain higher-order relations that are difficult to identify, thus representing a good testbed for our method.

**Ablation study.** We assess our core design choices on the real-world dataset as well. In Table 1 we are reporting the average displacement error (ADE) and final displacement error (FDE) metrics. Similar to the particle simulation datasets, we observe that predicting the hyperedges sequentially outperforms the simultaneous approach (*SPHINX w/o sequential* model) used by standard differentiable clustering algorithms. Moreover, the $k$-subset sampling proved to be clearly superior to the Gumbel-Softmax used in the previous methods (*SPHINX w gumbel* model).

**Comparison to recent methods.** In Table 3 we compare against other structure-based methods in the literature including graph-based methods: NRI (Fetaya et al., 2018), DNRI (Graber & Schwing, 2020), EvolveGraph (Li et al., 2020), STGAT (Huang et al., 2019), Trajectron++ (Salzmann et al., 2020) and hypergraph-based methods: higher-order-only EvolveHypergraph (H), the full EvolveHyper-

Table 2: **Performance on the transductive datasets** using various features. SPHINX obtains better results compared to both the static methods, where the structure doesn't modify during training, and dynamic methods, where the hypergraph is optimised. Mean/std computed over 20 runs.

|  | MODEL | NTU GVCNN | NTU MVCNN | NTU GV+MV | MODELNET40 GVCNN | MODELNET40 MVCNN | MODELNET40 GV+MV |
|---|---|---|---|---|---|---|---|
| STATIC | GCN | 78.80±0.92 | 78.72±1.97 | 80.43±1.09 | 91.80±0.46 | 91.50±1.80 | 94.85±1.75 |
| | GAT | 79.60±0.03 | 78.50±1.17 | 80.16±1.08 | 91.65±0.25 | 90.07±0.41 | 95.75±0.14 |
| | HGNN | 82.50±1.62 | 79.10±0.90 | 83.64±0.37 | 91.80±1.73 | 91.00±0.66 | 96.96±1.43 |
| | HGNN+ | 82.80±1.11 | 76.40±1.17 | 84.18±0.82 | 92.50±0.08 | 90.60±1.68 | 96.92±1.81 |
| | HNHN | 83.10±1.89 | 79.60±0.79 | 80.60±0.95 | 92.10±1.76 | 91.10±1.84 | 93.80±1.84 |
| | HYPERGCN | 79.90±1.78 | 78.10±0.83 | 79.90±0.91 | 92.20±0.80 | 90.20±0.28 | 96.10±0.63 |
| DYNAMIC | DHGNN | 82.30±0.98 | 77.60±1.55 | 85.13±0.26 | 92.13±1.55 | 85.53±0.83 | 96.99±1.46 |
| | DEEPHGSL | 76.28±1.45 | 72.30±1.57 | 78.67±0.77 | 89.32±0.71 | 88.62±0.93 | 90.33±0.66 |
| | HSL | 81.82±1.30 | 75.68±1.41 | 82.26±1.20 | 93.17±0.25 | 91.44±0.42 | 96.92±0.41 |
| | TDHNN* | 92.70±1.61 | 90.76±1.05 | 92.70±1.26 | 96.96±0.34 | 91.92±0.50 | 98.69±0.30 |
| | SPHINX | **94.80±0.90** | **92.95±0.85** | **94.67±0.71** | **97.29±0.29** | **92.79±0.41** | **98.92±0.21** |

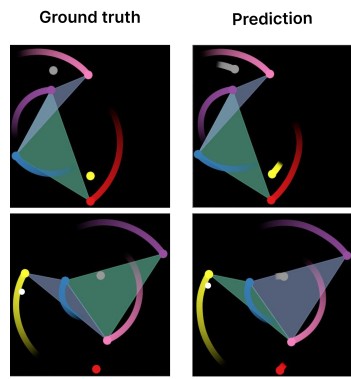

Figure 3: **Predicted trajectories and the latent hypergraph** on two examples of the Particle Simulation.

graph (H+G) (Li et al., 2022), GroupNet (Xu et al., 2022) and TDHNN (Zhou et al., 2023). Our method improves the short and medium-term prediction, while obtaining competitive results on the long-term prediction. Moreover, it is the only hypergraph-based method that does not require any auxiliary loss function or additional pairwise predictions.

The metrics used in the recent literature (minADE$_{20}$ and minFDE$_{20}$) take into account only the best trajectory from a pool of 20 sampled trajectories. Recently (Greer et al., 2020) show that these metrics suffer from a series of limitations, favouring methods that produce diverse, but less accurate trajectories. Thus, in Table 1, we also compared against the existing hypergraph-based methods using the single-trajectory prediction metrics (ADE/FDE) (the code for EvolveHypergraph is not publicly available). The results clearly show that for single-trajectory prediction our method obtains better results on both short and long-term prediction.

**Implementation details.** In all experiments we use the split from (Xu et al., 2022). For the observed trajectory the features are enhanced with the velocity. For the unobserved trajectory, the velocity is computed based on the predicted position. The models are trained for 300 epochs, using Adam with lr. 0.001, decreased by a factor of 10 when reaching a plateau. We treat the sampling algorithm as a hyperparameter, experimenting with AIMLE and SIMPLE.

### 4.3. Transductive setup

Most of the hypergraph inference methods are constrained to only predict a single hypergraph, which makes them unsuitable for the inductive setup. To allow direct comparison with all methods, we perform experiments on two transductive datasets for object classification. ModelNet40 (Wu et al., 2014) contains 12311 objects of 40 types, while NTU (Chen et al., 2003) contains 2012 objects with 67 types.

**Comparison to recent methods.** We compare against both static methods (where the structure does not adapt to the task) GCN, GAT (Veličković et al., 2018), HGNN, HGNN+ (Gao et al., 2023), HNHN (Dong et al., 2020), HyperGCN (Yadati et al., 2019) and dynamic hyperaph methods (where the structure is learned) DHGNN (Jiang et al., 2019), DeepHGSL (Zizhao et al., 2022), HSL (Cai et al., 2022), TDHNN. Table 2 shows that SPHINX learns a better latent structure, improving the final performance.

Note that, upon retraining the TDHNN model, we improved their official results. For fairness we report the improved performance (*TDHNN\**). However, qualitative evaluation showed that in all cases each hyperedge learned by TDHNN ends up containing all the nodes in the dataset. While the model still obtains a good downstream performance, it seems that this is the results of learning good attention coefficients rather than learning a useful sparse structure.

**Implementation details.** For all transductive experiments we adopt the split from (Zhou et al., 2023) and train the model with three types of visual features: GVCNN (Feng et al., 2018), MVCNN (Su et al., 2015) or combining both of them. We are treating the number of slots and the cardinality of slots as hyperparameters. Unless otherwise stated, we use AllDeepSet model as hypergraph processor.

## 5. Conclusion

We are introducing SPHINX, a model for unsupervised hypergraph inference that enables higher-order processing in tasks where hypergraph structure is not provided. The model adopts a global processing in the form of sequential soft clustering for predicting hyperedge probability, paired with a $k$-subset sampling algorithm for discretizing the structure such that it can be used with any hypergraph neural network decoder and eliminating the need for heavy optimisation tricks. Overall, the resulting method is a general and broadly applicable solution for relational processing, facilitating seamless integration into various real-world systems perceived to necessitate higher-order analysis.

**Acknowledgment** The authors would like to thank Charlotte Magister, Daniel McFadyen, Alex Norcliffe and Paul Scherer for fruitful discussions and constructive suggestions during the development of the paper.

## Impact Statement

This paper presents work whose goal is to advance the field of Machine Learning. There are many potential societal consequences of our work, none which we feel must be specifically highlighted here.

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

# Appendix

This appendix contains details related to our proposed Structural Prediction using Hypergraph Inference Network (SPHINX) model, including broader impact and potential limitations, qualitative visualisations from our model and the baselines we compared against in this paper, details about the proposed dataset, information about the implementation of the model and additional experiments referred in the main paper. We will soon release the full code associated with the paper.

- **Section A** highlights a series of potential limitations that can be address to improve the current work, together with a discussion about the social impact of our approach.

- **Section B** presents an overview of the existing methods for hypergraph prediction, with a focus on a few qualities that we believe a general methods should fulfill.

- **Section C** present the computational complexity of our model.

- **Section D** contains visualisation of the discovered hypergraph structure and the predicted trajectory for our proposed model and different variations of it.

- **Section E** provides more details about the proposed synthetic dataset, further information about the method and its training process.

- **Section F** presents additional ablation studies showing the influence of varying the number of hyperedges in our model, but also a more detailed analysis of the synthetic experiments presented in the paper.

## A. Broader Impact & Limitations

In this paper, we are proposing a framework for processing higher-order interactions in scenarios where a ground-truth hypergraph connectivity is either too expensive or not accessible at all. We are showing that our model is capable of recovering the latent higher-order structure without supervision at the hypergraph level, thus improving the final, higher-level performance. While we mainly tested on trajectory prediction tasks, our model is general and versatile and can be applied to any scenarios where we suspect capturing and modeling higher-order relations can be beneficial. Consequently, we believe that our model doesn't have any direct negative societal impact. Moreover, the method has the advantage of allowing us to inspect the learned latent hypergraph structure (as we can see in Section D of this Supplementary Material). This offer an advantage in terms of interpretability, allowing us to better understand and address potential mistakes in the model's prediction.

By construction, our model predicts a set of $M$ slots, each one corresponding to a hyperedge. This process is sequential, with one slot predicted at each time-step. While stopping criterion can be imposed to allow dynamic number of hyperedges, in order to be able to take advantage of the historical features $b \in \{0,1\}^{N \times (M-1)}$ we need to have access to a pre-defined maximum number of hyperedges $M$. In the current version of our work, this number is picked as a hyperparameter. However, in Section D, we experimentally show that the chosen value doesn't have a significant impact on the results. As long as the number of hyperedges is larger than the expected one, the performance doesn't decrease.

Furthermore, as we show in our experimental analysis, the $k$-subset sampling has a clear, positive impact on the performance of our model. However, this comes with a limitation: the sampled hyperedge is constrained to have exactly $k$ nodes. During training, this constrain helps the optimisation in terms of stability, allowing us to easily train our model without the need for regularisation or optimisation tricks. However, we are restricted to predict only $k$-regular hyperedges (i.e. hyperedges containing exactly $k$ nodes). While in all our experiments we use the same constant $k$ for all hyperedges, the framework allows us to use a sequence of cardinalities, ones for each hyperedge. However, additional domain knowledge is needed to pre-determined this sequence. Since having extra slots proved to be harmless for our model, one solution is to provide a diverse list of $k$ such that it covers a broader range of possibilities. While interesting to explore, these experiments are left as future work.

On the inductive setup, in order to better align with the current literature, we mainly validate our model on the trajectory prediction task. However, in the current form, our model predicts a single hypergraph structure for the entire trajectory. This is well suited for the synthetic benchmark, where the dynamics for each example is determined by a single hypergraph structure, that does not change in time. On the other hand, in real-world setups, such as the NBA dataset, the higher-order interactions between the players might change along the game. Thus, having a distinct, dynamic hypergraph structure, that

evolves in time, could be beneficial. However, solving trajectory prediction is not the main goal for our model. Instead, our purpose is to have a general model, that allows us to infer and process higher-order interaction. We believe that adapting SPHINX to more specific scenarios, such as creating a dynamic, evolving structure is an interesting area for future work.

## B. Overview of the existing dynamic hypergraph predictors

Table 4: Overview of the existing methods for hypergraph prediction.

| METHOD | INDUCTIVE | TRANSDUCTIVE | DIFFERENTIABLE | DISCRETE | REGULARISATION-FREE | |
|---|---|---|---|---|---|---|
| DEEPHGSL (ZIZHAO ET AL., 2022) | ✓ | ✓ | ∼ | ✓ | ✓ | REQUIRES INITIAL HGRAPH |
| MHSL (LEI ET AL., 2024) | ✓ | ✓ | ∼ | ✗ | ✓ | REQUIRES INITIAL HGRAPH |
| HERALD (ZHANG ET AL., 2022) | ✓ | ✓ | ✓ | ✗ | ✓ | REQUIRES INITIAL HGRAPHS |
| ADA-HYPER (YU ET AL., 2012) | ✗ | ✓ | ✓ | ✗ | ✗ | |
| DYHL (ZHANG ET AL., 2018) | ✗ | ✓ | ✗ | ✓ | ✗ | |
| DHGNN (JIANG ET AL., 2019) | ✗ | ✓ | ✗ | ✓ | ✓ | |
| (D)HSL (WANG ET AL., 2024; CAI ET AL., 2022) | ✗ | ✓ | ✓ | ✓ | ✓ | |
| TDHNN (ZHOU ET AL., 2023) | ✓ | ✓ | ∼ | ✗ | ✗ | |
| (DYN)GROUPNET (XU ET AL., 2022; 2024) | ✓ | ✓ | ∼ | ✓ | ✗ | |
| EVOLVEHYPERGRAPH (LI ET AL., 2022) | ✓ | ✓ | ✓ | ✓ | ✗ | |
| **SPHINX (OURS)** | ✓ | ✓ | ✓ | ✓ | ✓ | |

For a better understanding of the field, we are offering an overview of the existing methods for dynamic hypergraph prediction, focusing on the three desiderata that we highlight in the introduction: 1) **The model should be applicable on a broad types of tasks.** For a general hypergraph prediction method, it should be adaptable for both the inductive and the transductive setup. 2) **The model should be compatible with any hypergraph processing architecture.** This requires the model to be differentiable and to produce a discrete output such that it can be easily plugged into any hypergraph decoder. 3) **A powerful model should be easy to optimise.** Most of the models requires additional regularisation terms or residual branches in order to optimise the hypergraph predictor well. Ideally, the model should allow to incorporate the hypergraph predictor component without negatively impacting the optimisation process.

For completeness, we are including also methods such as DeepHGSL, MHSL and HERALD, but these methods does not directly generate a hypergraph structure, but rather rewire an existing one. As we can see in Table 4, half of the models are unapplicable in the inductive setup mainly due to the fact that they are treating the structure as a parameter. The models marked with the symbol ∼ are only partially differentiable. Models such as top-k or threasholding, when using to infer a discrete structure, loses the gradient. Moreover, the table also suggests that most of the methods are having issue in optimising a sparse, discrete structure without incorporating various regularisation constraints.

The results presented in our main paper confirm the fact that SPHINX is a general and easy to adapt method, as it obtains good performance across various setups, without any additional regularisation.

## C. Computational complexity.

We analyse the computational complexity of the hypergraph-inference component for the SPHINX architecture (hypergraph predictor + sampling) when using the SIMPLE $k$-subset sampling algorithm. The overall complexity would be obtained by adding the complexity specific to the hypergraph neural network used to process the higher-order structure. If $N$ is the number of nodes, $K$ is the cardinality of the predicted hyperedges and $M$ is the number of inferred hyperedges, the vectorized complexity for the sampling module is $O(logN \times logK)$ and the complexity of slot-attention with $Q$ iterations and one slot is $O(Q \times N)$. Then, the overall complexity for predicting $M$ hyperedges is $O(M \times Q \times N + M \times logN \times logK)$.

## D. Visualisations

One of the advantages brought by our method is providing a discrete latent hypergraph structure, that can be easily visualise, offering a higher level of interpretability compared to attention-based methods. Figure 6 shows the trajectories and latent structure learned by our model on some examples from the Particle Simulation and NBA datasets. We are visualising the ground-truth trajectories, the trajectories and structure predicted by our full model, and the trajectories and learned structure corresponding to the model used in our ablation study: *SPHINX w/o sequential* and *SPHINX w Gumbel*.

When the slots are predicted in parallel, without access to the already predicted hyperedges (the model called *SPHINX w/o*

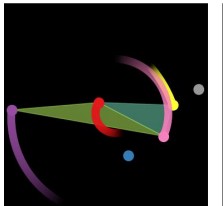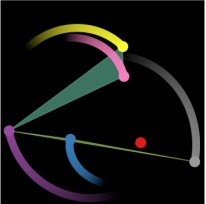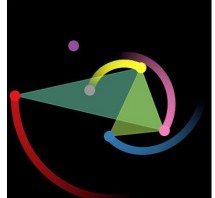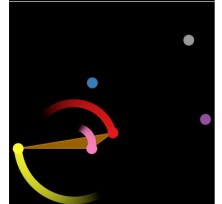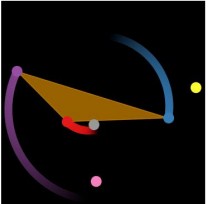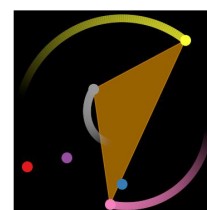

Figure 4: **Examples of** $25$**-steps trajectories from the Particle Simulation datasets**. The highlighted triangles represent the 3-order interactions used to generate the trajectories.

*sequential*), the slots tends to collapse in a single hyperedge represented by all the slots. On the other hand, removing the constraints on the cardinality of the hyperedges, by applying Gumbel-Softmax (the model called *SPHINX w Gumbel*) , the optimisation process leads to "greedy" slots, that covers all the nodes in the hypergraph.

In contrast, our *SPHINX* model learns sparse and intuitive hypergraph structures, alleviating the slot collapsing behaviour. On the synthetic dataset, the predicted hypergraph highly coincide with the real connectivity. By inspecting the higher-order structures learned by our model on the NBA dataset, we observe that most of the inferred hyperedges contains the node associated with the ball (denoted as a white node in Figure 6). we believe that this is a natural choice, since the position of the ball should have a critical influence on the decision taken by all basketball players.

**Insights on the inferred hypergraph.** Evaluating the accuracy of predicted latent hypergraphs in real-world data is a critical but challenging problem, that remains an open problem. Our datasets lack annotations for higher-order structures, and even if they would have existed, there's no guarantee they would be optimal for the task.

However, to understand the type of hypergraph predicted by our model in the real-world datasets, we analyze the NBA dataset, an inductive dataset that is easier to visualize and interpret. For a model with 6 hyperedges and 4 nodes per hyperedge:

- the average node degree for the predicted hypergraphs are $[1.97, 1.98, 1.99, 1.99, 1.98, 1.95, 1.96, 1.96, 1.96, 1.95, 4.29]$. The first $10$ nodes correspond to players- model focuses uniformly on all of them; while the last node represents the ball- the model learns to include the ball in most hyperedges. Thus the predicted structures are star-like hypergraphs, aligning with our intuition that player movement is highly influenced by the ball's position.

- the average percentage of duplicated hyperedges across examples is $2.74\%$.

This shows a high diversity in our predictions, confirming that the model captures not just the most likely connections but also a scene-specific structure.

## E. Experimental details

### E.1. Datasets details

We create a synthetic benchmark, Particle Simulation, to validate the capabilities of our model in terms of downstream performance, while also enabling the evaluation of the predicted higher-order interactions. Previous works used various synthetic setups to achieve this goal. However, to our knowledge, none of them are publicly available.

Particle Simulation dataset contains a set of particles moving according to a higher-order law. Each example in our dataset consists of $N = 6$ particles moving in a 2D space. Among them, $K$ random triangles where uniformly sampled to represent $K$ 3-order interactions. All the particles that are part of a triangles are rotating around the triangle's center of mass with an angular velocity $\theta_i$ randomly sampled for each particle $i$. If one particle is part of multiple triangles, the rotation will happen around their average center of mass. The task is, given the first $t = 22$ time steps of the trajectory (only the position of the particles and the angle velocity associated with the particles, no triangles provided), to be able to predict the rest of the trajectory (25 steps). Note that, for each example in the dataset, we have different triangles connecting the particles. Given the hypergraph structure, the task satisfy the Markovian property, but in order to predict the hypergraph structure you need access to at least 2 consecutive timesteps.

The dataset contains two higher-order interaction per trajectory. The dataset split constitutes of 1000 trajectories for training, 1000 forvalidation and 1000 for test.

Some examples of the dataset are depicted in Figure 4.

### E.2. Implementation details

**Model details.** To encode the observed part of the trajectory into the hypergraph predictor, we are experimenting with two variants. Either we are using an MLP that receives the concatenation of the node's coordinates from all timesteps, or a 1D temporal Convolutional Neural Network acting on the temporal dimension.

For the synthetic dataset, both the hypergraph predictor and the hypergraph processor are receiving as input the (x,y) coordinates of each node, while for the NBA dataset the spatial coordinates are enriched with information about the velocity (the real velocity for the observed trajectory and the estimated one during evaluation).

For the AllDeepSets model, used as Hypergraph Processor, we adapt the standard architecture, by incorporating information about the angular velocity $_i$ at each layer, in the form of $sin(\theta_i)$, $cos(\theta_i)$ concatenated to each node at the end of each layer.

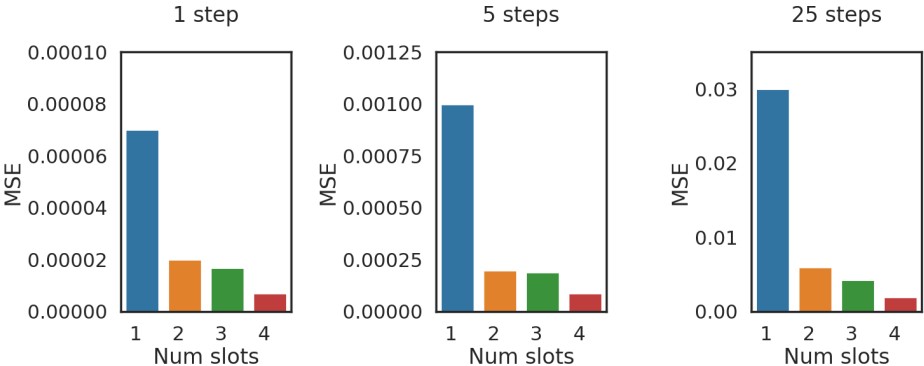

Figure 5: **Experiments on Particle Simulation dataset investigating the importance of choosing the correct number of predicted hyperedges**. Having less hyperedges than the real number (two) negatively impact the performance. However, when increasing the number of hyperedges above the true value, the performance does not deteriorate, indicating that the model is robust at discovering the hypergraph structure, as long as enough slots are provided.

Unless otherwise specified, the results reported in the main paper are obtained using hyper-parameter tuning. We are performing bayesian hyperparameter tuning, setting the base learning rate at $0.001$, multiplied with a factor of d $\in$ $\{0.1, 1.0, 10.0\}$ when learning the parameters corresponding to the hypergraph predictor, a batch size of $128$, self-loop added into the structure. The hidden state is picked from the set of values $\{32, 64, 128, 256\}$, the number of AllDeepSets layers from $\{1, 2\}$, the number of layers for the used MLPs from $\{2, 3, 4\}$, the number of hyperedges from $\{1, 2, 3, 5, 7\}$ (except for the synthetic setup where the number of hyperedges is set to 1 and 2 respectively), the dimension of hyperedges from $\{3, 4, 5, 6\}$ (except for the synthetic setup where the number of hyperedges is set to 3), nonlinearities used for the similarity score are either sigmoid, sparsemax or softmax, the algorithm for $k$-subset sampling is either AIMLE or SIMPLE, with their associated noise distribution sum of gamma or gumbel. For the NBA dataset, we are training for 300 epochs, while for the Particle Simulation we are training for 1000 epochs.

The code for loading the NBA dataset is based on the official code for (Xu et al., 2022)[4]. The code for the various $k$-subset sampling algorithms is based on the code associated with AIMLE[5] (Minervini et al., 2023), IMLE[6] (Niepert et al., 2021) and SIMPLE[7] (Ahmed et al., 2023) methods.

**Metric details.** To validate to what extent our model is able to predict the correct hypergraph structure, we are computing the overlap between the predicted hypergraph and the ground-truth connectivity used to generate the trajectory. For a model

---

[4]https://github.com/MediaBrain-SJTU/GroupNet - under MIT License

[5]https://github.com/EdinburghNLP/torch-adaptive-imle - under MIT License

[6]https://github.com/uclnlp/torch-imle under MIT License

[7]https://github.com/UCLA-StarAI/SIMPLE - custom License, open for research

predicting $K$ hyperedges, let's consider the set of predicted hyperedges as $P = \{p_1, p_2..p_K\}$ and the target hyperedges as $G = \{g_1, g_2..g_K\}$. For each element $i$, we are computing the percentage of the nodes that are in both the prediction $p_i$ and the ground-truth hyperedge $g_i$. Since we are computing the element-wise overlap between two unordered sets, we fix the order of the ground-truth list $G$, and compute the metric against all the possible permutations of $P$. The overlap corresponding to the best permutation represents our reported metric. This metric is a scalar between $0$ and $1$, with $0$ representing completely non-overlapping sets, and $1$ denoting perfect matching.

# F. Additional experiments

## F.1. Description of the models used in Particle Simulation

In the following, we will describe all the baselines used in the Particle Simulations ablation study.

*Random structure* Instead of predicting the hypergraph structure, this model uses a randomly sampled hypergraph connectivity. The goal is to understand how much of our performance comes from having a strong decoder, compared to the advantage of predicting a real, suitable hypergraph structure.

*No structure* entirely ignores the higher-order interaction, by setting the incidence matrix to zero. The purpose of this baseline is to understand to what extent our datasets require higher-order interactions.

*SPHINX w/o sequential* uses the same processing as our full proposed model. However, instead of a sequential slot-attention, it uses the classical slot-attention clustering, which predicts all the slots simultaneously. Comparison with this baseline sheds light on the importance of predicting the hyperedges sequentially.

*SPHINX w Gumbel* is a modification of our full SPHINX model, by replacing the $k$-subset sampling with the Gumbel-Softmax, frequently used in both graph and hypergraph inference models. While allowing us to propagate gradients through the sampling step, the Gumbel-Softmax has no constraints on the number of nodes contained in each hyperedge, sampling independently for each node-hyperedge incidence relationship.

*TDHNN and GroupNet* We adapt the models proposed in (Zhou et al., 2023) and (Xu et al., 2022) respectively to predict the group of particles that are interacting in the Particle Simulation dataset. To ensure a fair comparison we are using the same decoder in all experiments and only modify the encoder.

## F.2. Varying. number of hyperedges

Although the number of parameters does not scale with the number of inferred hyperedges, by construction we are restricted to predict maximum $M$ hyperedges, where $M$ is set as hyperparameter. This is mainly due to the use of binary feature vector $b$ encoding the history predicted so far, which has a fixed dimension $\{0, 1\}^{N \times (M-1)}$.

To establish to what extent the value we choose as the number of hyperedges impacts the performance, we conduct an ablation study by varying the number of slots predicted by our method. The results in Figure 5 show that picking a number of slots larger than the real one doesn't harm the performance. Visual analysis of the learned hypergraph reveals that, when offering more slots than needed, the model tends to produce redundancy, associating the extra slots to hyperedges that were already discovered. To better understand this behaviour, we measured the average number of distinct hyperedges predicted when providing more slots than required. The results are show in Table 5.

Table 5: **Ablation study providing more slots than the real number.** *Unique Hyperedges* denotes the number of unique predicted hyperedges while *MSE* denotes the final performance.

| Slots | Unique Hyperedges | MSE |
|---|---|---|
| 1 slot | 1.00 | 0.00007 |
| 2 slots | 1.96 | 0.00002 |
| 3 slots | 2.15 | 0.000017 |
| 4 slots | 2.14 | 0.000007 |

We believe that this is a very interesting result, as it shows that when equipped with more slots than necessary the model learns some form of redundancy (by predicting the same hyperedge multiple times) instead of hallucinating fake relations.

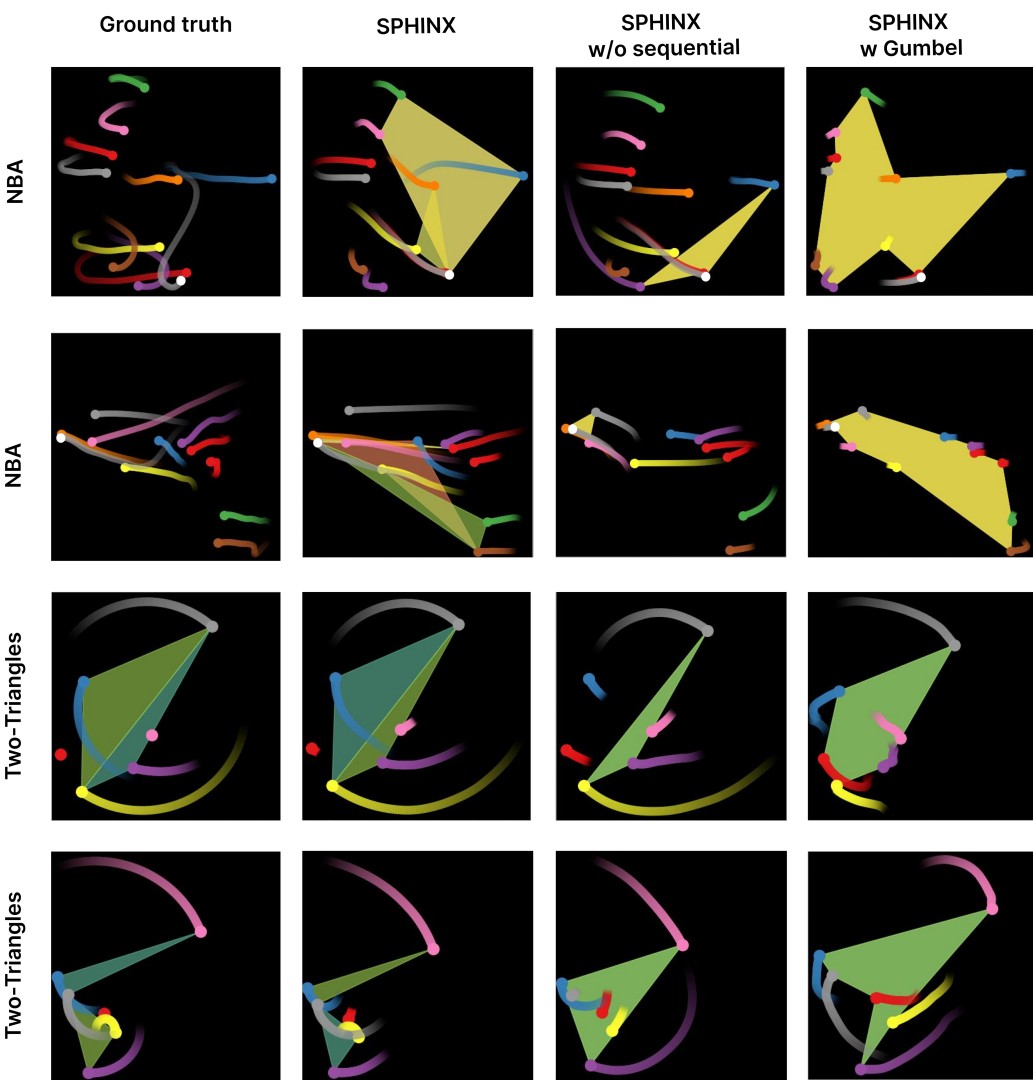

Figure 6: **Visualisation of the trajectories** predicted by our full model (SPHINX) and the two variations used in the ablation study (SPHINX w/o sequential and SPHINX w Gumbel) on the Particle Simulation and NBA dataset. In the ground-truth column the highlighted polygons represent the true connectivity, while for the models they represent the discovered hyperedges. Both using Gumbel-Softmax and dropping the sequential prediction clearly impact the predicted hypergraphs. Gumbel-Softmax struggles to produce a sparse structure, while the lack of sequentiality leads to both predicted hyperedges containing the same set of nodes. On the other hand, our model manage to discover diverse structures, close to the ground-truth ones in the synthetic scenario.

