# OpenReview forum: "SPHINX: Structural Prediction using Hypergraph Inference Network"
_ICML.cc/2025/Conference — ICML 2025 poster_

### Official Review · Reviewer_HCWJ · 2025-03-08

**Overall Recommendation:** 4

**Summary:**

SPHINX proposes an unsupervised framework for latent hypergraph inference that models higher-order interactions directly from point-wise data. The method employs a sequential slot attention mechanism to predict hyperedge probabilities and uses differentiable k-subset sampling to convert these probabilities into a discrete hypergraph structure. This latent structure is then integrated into any hypergraph neural network, enabling improved performance on both inductive and transductive tasks, as demonstrated through extensive experiments and ablation studies.

**Claims And Evidence:**

The paper’s claims are supported by clear and convincing empirical evidence. Extensive experiments show improvements in hypergraph accuracy and downstream task performance, effectively validate the approach.

**Essential References Not Discussed:**

The paper cites most of the key works in hypergraph learning and unsupervised relational inference.

**Experimental Designs Or Analyses:**

The experimental design is robust and comprehensive.

**Methods And Evaluation Criteria:**

The proposed methods—sequential slot attention for hyperedge inference and differentiable k-subset sampling for discretization—are well-motivated and appropriate for the challenge of inferring latent hypergraph structures. I especially love the fact that SPHINX can be used with any hypergraph neural network decoder and eliminating the need for heavy optimisation tricks.

**Other Comments Or Suggestions:**

NA

**Other Strengths And Weaknesses:**

I think the hyperparameter M is a really strong prior and needs to be carefully tuned. It would be better if there is some way to select M (or heuristically set an appropriate M).

**Questions For Authors:**

NA

**Relation To Broader Scientific Literature:**

The pipeline could be inspiring for other graph structure learning methods.

**Theoretical Claims:**

NA

---

> ### Author Rebuttal · Authors · 2025-03-31
>
> We appreciate the reviewer’s positive feedback on our work. Below, we address the key points raised and we will incorporate all suggestions into the final version of the paper.
>
> **Selecting the number of hyperedges M**
>
> Having a fixed maximum number of hyperedges is a limitation that we are sharing with most of the hypergraph predictor methods in the literature  and we agree with the reviewer that having a fully dynamic model, allowing dynamic number of hyperedges, is an important direction.
>
> The experiments presented in the Appendix (Fig 5)  are designed to understand to what extent this affects the performance of the model on the synthetic setup (where we know what is the real number of hyperedges needed). The results show that, as expected, having a too small M causes a drop in performance while having an M that is larger than the golden standard does not affect the performance.
> To better understad this behaviour, we measured the average number of distinct hyperedges predicted when providing more slots than required. The results are as follow:
>
> | Slots      | Unique Hyperedges | MSE       |
> |------------|------------------|-----------|
> | **1 slot**  | 1.00             | 0.00007   |
> | **2 slots** | 1.96             | 0.00002   |
> | **3 slots** | 2.15             | 0.000017  |
> | **4 slots** | 2.14             | 0.000007  |
>
> We believe that this is a very interesting result, as it shows that when equipped with more slots than necessary the model learns some form of redundancy (by predicting the same hyperedge multiple times) instead of hallucinating fake relations.

---

> > ### Comment · Reviewer_HCWJ · 2025-04-05
> >
> > Thanks for adding hyperparameter analysis for $M$. I would like to keep my positive score.

---

### Official Review · Reviewer_BF7S · 2025-03-12

**Overall Recommendation:** 4

**Summary:**

The authors focus on unsupervised hypergraph inference. Three key desiderata are identified: applicability to a broad type of tasks, compatibility with any hypergraph process architecture, and ease of optimization. They propose SPHINX that adapts the slot attention for sequential hyperedge prediction, and is differentiable by leveraging contrained k-subset sampling. SPHINX is compatible with popular hypergraph neural networks. Experiments are conducted on synthetic and real-world datasets. Extensive results and analysis verify the superiority of SPHINX, and how its key components take effect.

**Update after rebuttal**

I have read the response from the authors. Since my original recommendation was to accept, I will maintain my recommendation.

**Claims And Evidence:**

The claim in contributions is supported by experimental results, including accurate hypergraph discovery and how it benefits downstream tasks.

**Essential References Not Discussed:**

A previous that uses sequential prediction for graph generation is not reviewed.

Efficient Graph Generation with Graph Recurrent Attention Networks. In NeurIPS, 2019.

**Experimental Designs Or Analyses:**

The experimental design is rather comprehensive.
1. Experiments are conducted on both synthetic datasets and real-world datasets. Personally, I think it is creative to design the particle simulation to verify the accuracy of hypergraph discovery.
2. SPHINX is compared with representative baselines and state-of-the-art methods
3. Hypergraph inference is studied under both the inductive and transductive settings.
4. Various ablation studies are conducted to validate the key features of SPHINX, e.g., sequential prediction, k-sampling, etc.

**Methods And Evaluation Criteria:**

1. SPHINX is designed under the guidance of three desiderata pointed out by the authors. SPHINX is well-motivated and reasonably designed.

2. Clustering-based hyperedge discovery and k-subset sampling to allow backpropagation are reasonable for end-to-end hypergraph inference.

3. Designing particle simulation to verify the effectiveness of SPHINX on hypergraph prediction is reasonable because the ground truth are known.

4. The transductive and inductive settings for hypergraph inference are widely accepted.

**Other Comments Or Suggestions:**

Personally, I think more technical descriptions of k-subset sampling, a key technique in SPHINX, should be added, either in the main text or the appendices.

**Other Strengths And Weaknesses:**

Strengths:
S1. SPHINX is technically solid and empirically significant under variant experimental settings.

S2. It is innovative to apply the slot attention in computer vision to hypergraph clustering.

Weaknesses:

W1. SPHINX may not be general enough to infer arbitrarily structured hypergraphs. Please see Q1 for details.

**Questions For Authors:**

Q1. I have several questions on hypergraph inference from the algorithmic perspective.

Q2. How to determine the number of total hyperedges M, and the size of each hyperedge k, i.e., the number of nodes it contains. Are they specified before hypergraph inference? How to decide the best M and k? Specifically, M and the largest k reflect the intrinsic order of higher-order interaction. Thus, I believe it is crucial for understanding the complexity of the interaction.

Q3. Does your algorithm allow hyperedges of different sizes?  Besides, I note that the authors analyze the computational complexity in Appendix C. Is it possible to report the running time?

Q4. How do you determine the order of sequential prediction for hyperedges? Will that affect the performance?

Q5. Can you analyze the inferred hypergraphs on real-world datasets? It will help the readers understand the meaning the hyperedges and better justify the contributions of ``adding a new layer of interpretability to the model''. The authors may consider reporting some statistics of the hypergraphs (e.g., number of hyperedges and the largest size of hyperedges) and conducting some case studies.

**Relation To Broader Scientific Literature:**

This paper is related to structural inference, higher-order network learning, hypergraph neural networks, etc.

**Theoretical Claims:**

N/A

---

> ### Author Rebuttal · Authors · 2025-03-31
>
> We appreciate the reviewer’s comments and feedback. We thank the reviewer for pointing out Liao et al, which shares similarities with our work in sequential structure prediction and offers inspiration for future improvements. We will include a discussion in the paper.
>
> **Different hyperedge sizes**
>
> SPHINX allows hyperedges of different sizes. The argument k in the sampling algorithm does not affect the number of learnable parameters so it can vary among hyperedges. For simplicity and to reduce hyperparameter search space, we fix k across all hyperedges. However, if a more diverse hypergraph is needed, an array of k-values can be given as input.
>
> **Choosing M and k**
>
> As mentioned in the paper, all experiments treats M and k as hyperparameters.
>
> Dynamic hyperedge sizes: We agree that a dynamic k could enhance flexibility and is a promising direction for improvement. An alternative to hand-picking k is to select it according to the probability distribution: defining k as the number of nodes above a probability threshold or the rank with a significant gap in the distribution. Though still non-differentiable, this removes the need for predefined hyperedge sizes.
>
> Sensitivity to the number of hyperedges M: Having a fixed maximum number of hyperedges is a limitation that we share with most hypergraph predictors in the literature. The experiments in Fig 5 examine how this impacts performance in the synthetic setup. As expected, a too small M causes a drop in performance while an M that is larger than the oracle(2) does not affect the performance. To understand this behaviour, we measured the average number of unique hyperedges. Results show that, when equipped with more slots than necessary, the model learns a form of redundancy (predicting the same hyperedge multiple times) instead of hallucinating fake relations.
> |Slots|Unique Hedges|MSE|
> |-|-|-|
> |**1** |1.00|0.00007|
> |**2**|1.96|0.00002|
> |**3**|2.15|0.000017|
> |**4**|2.14|0.000007|
>
> That said, we agree that developing a fully dynamic model with adaptive k and M is an important future direction.
>
> **Insights on the inferred hypergraph**
>
> Evaluating the accuracy of predicted latent hypergraphs in real-world data is a critical but challenging problem, that remains an open problem. Our datasets lack annotations for higher-order structures, and even if they would have existed, there’s no guarantee they would be optimal for the task.
>
> However, to understand the type of hypergraph predicted by our model in the real-world datasets, we analyze the NBA dataset, an inductive dataset that is easier to visualize and interpret. For a model with 6 hyperedges and 4 nodes per hyperedge:
> - the avg node degree for the predicted hypergraphs are [1.97, 1.98, 1.99, 1.99, 1.98, 1.95, 1.96, 1.96, 1.96, 1.95, 4.29]. The first 10 nodes correspond to players- model focuses uniformly on all of them; while the last node represents the ball- the model learns to include the ball in most hyperedges. Thus the predicted structures are star-like hypergraphs, aligning with our intuition that player movement is highly influenced by the ball’s position.
> - the avg percentage of duplicated hyperedges across examples is 2.74%. This shows a high diversity in our predictions, confirming that the model captures not just the most likely connections but also a scene-specific structure.
>
> **Description of  k-subset sampling**
>
> To maintain readability, we initially omitted certain technical details, such as the formulation of k-sampling. However, we agree that including this information will make the paper more self-contained and reproducible. We will add a section in the appendix.
>
> **Running time**
>
> Following your suggestions, we measured time per iteration on NBA(batch size 128, 11 nodes) and ModelNet(12311 nodes) using a Quadro RTX 8000 GPU. While ensuring architectural comparability, our code is not optimized for large-scale data. More engineering improvements can further enhance scalability.
> ||ModelNet| | |NBA| | |
> |-|-|-|-|-|-|-|
> ||SPHINX |TDHNN||SPHINX|TDHNN|GroupNET|
> |**Training Time (sec)**|0.32 |0.77||0.018|0.21|0.071|
> |**Infer Time (sec)**|0.26|0.76||0.030 |0.22|0.030|
>
> **Order of sequential prediction for hyperedges**
>
> We thank the reviewer for pointing this out. Since we do not have supervision at the hypergraph level, we believe order ambiguity is less pronounced than in graph-generation tasks (e.g., GRAN). But, sequential prediction still introduces equivalence classes in the latent space, where any predicted orders yield the same hypergraph, potentially causing model confusion.
>
> To address this, we initialize slot attention deterministically, using the same random sequence for all examples, allowing the model to learn a canonical order if one exists. However, we agree that this is a naive way of imposing an order and we think that there is room for improvement in that regard. We thank the reviewer for identifying the GRAN paper which might offer inspiration for avoiding order-ambiguity.

---

> > ### Comment · Reviewer_BF7S · 2025-04-02
> >
> > Thanks for your thoughtful discussion. Ideally, the model's prediction is invariant w.r.t. the prediction order, or the predictions form an equivalence class. However, such property needs theoretical support. A canonical order may remove ambiguity in practice, but it is unclear if the order affects the optimality. Different practitioners may choose different orders. Thus, I think it is worthwhile to resolve this issue in the future, which will benefit both theoretical and practical aspects. On the whole, I will keep my evaluation.

---

### Official Review · Reviewer_rEAs · 2025-03-16

**Overall Recommendation:** 3

**Summary:**

This paper introduces the SPHINX model, which aims to infer a latent hypergraph structure suitable for the final task in an unsupervised manner from input features, to support higher-order relationship processing in the absence of a readily available hypergraph structure. The process is divided into three steps: First, the hypergraph predictor infers a latent hypergraph based on the input features. Next, a k-subset sampling algorithm is used to transform the obtained probability distribution into specific incidence relationships. Finally, the predicted hypergraph is applied to a standard hypergraph neural network to generate higher-order representations. Experimental results show that SPHINX not only excels in inferring latent hypergraphs but also effectively enhances the performance of downstream tasks in both inductive and transductive settings.

**Claims And Evidence:**

Some of the arguments in the paper are not sufficiently substantiated. Although the authors claim that SPHINX can infer latent hypergraphs that are highly consistent with the true higher-order structures, the validation on real-world datasets is not adequate. There is no direct evidence to prove the match between the inferred hypergraphs and the actual higher-order relationships; the authors only provide indirect proof through the improved performance of downstream tasks. Additionally, the paper claims that the “model is easy to optimize,” but it does not provide a detailed analysis of the convergence speed and stability of the optimization process, nor does it offer comparative experiments to support this claim. As a result, the persuasiveness of this claim is weak.

**Essential References Not Discussed:**

Some relevant works are not cited or discussed in the paper. For example, in the field of graph neural networks in recent years, there have been some works on the learning and prediction of dynamic graph structures. These methods share some similarities with hypergraph inference in terms of ideas, but they are not mentioned in the paper.

**Experimental Designs Or Analyses:**

The experimental design has some deficiencies, especially in the selection of datasets. Although both synthetic and real-world datasets are included, the diversity of the real-world datasets is insufficient, which limits the validation of the model's effectiveness on different types of data and tasks. This affects the comprehensive evaluation of the model's actual performance. Increasing the diversity of datasets is crucial for verifying the model's robustness and generalization ability.

**Methods And Evaluation Criteria:**

The proposed method is rational to some extent, yet its evaluation criteria are limited. In addition to focusing on the improved performance of downstream tasks, the assessment of the hypergraph predictor should also consider multiple aspects of the hypergraph structure itself, such as its rationality, sparsity, and interpretability. However, the paper mainly focuses on the performance of downstream tasks and lacks rich indicators to evaluate the quality of the hypergraph structure itself. This single evaluation method may not fully reflect the effectiveness and superiority of the method.

**Other Comments Or Suggestions:**

None

**Other Strengths And Weaknesses:**

Strengths:The paper proposes a novel unsupervised hypergraph inference model that is compatible with existing hypergraph neural network architectures and can enhance the performance of downstream tasks. Additionally, the design of the model is innovative, especially the use of the k-subset sampling algorithm to generate discrete hypergraph structures, which provides a new approach for hypergraph inference.
Weaknesses:In addition to the previously mentioned issues of insufficient evidence, limited evaluation criteria, and lack of theoretical analysis, the scalability of the model is also a concern. When dealing with large-scale datasets, the computational complexity of the hypergraph predictor may be high, which could affect the efficiency and practicality of the model.

**Questions For Authors:**

1.How can the consistency between the inferred hypergraph structure and the true higher-order relationships of the hypergraph structure be verified on real-world datasets? Are there any more direct validation methods other than the improved performance of downstream tasks?
2.In Section 3.1, the authors generate discrete hypergraph structures using a hypergraph predictor and a k-subset sampling algorithm. However, regarding the interpretability of the generated hypergraph structures, can the authors provide a more detailed analysis?
3.In Section 4, the authors demonstrate the performance of SPHINX on several benchmark datasets. However, regarding the computational cost when dealing with large-scale graph data, do the authors have any plans to optimize the algorithm to enhance its scalability?
4.In the experimental section, the authors primarily conducted experiments on synthetic datasets and a few common benchmark datasets. However, the diversity and complexity of these datasets are limited, which prevents a comprehensive validation of the method's effectiveness on different types of graph data and tasks.

**Relation To Broader Scientific Literature:**

The key contributions of the paper are related to the existing literature to some extent, but the elaboration is not deep enough. Hypergraph inference, as an emerging field, intersects with areas such as graph neural networks and structural prediction. However, when discussing related work, the authors mainly focus on recent studies and lack sufficient review of classical theories and methods. This makes the background introduction of the paper not comprehensive enough, making it difficult to fully demonstrate its innovativeness and significance.

**Theoretical Claims:**

The paper lacks sufficient theoretical analysis regarding the model's generalization ability and its applicability across different data distributions, which undermines the robustness of the model's theoretical foundation.

---

> ### Author Rebuttal · Authors · 2025-03-31
>
> We sincerely appreciate your detailed review and valuable feedback on our paper. We would like to address your concerns and questions.
>
> **Hypergraph evaluation for real-world datasets**
>
> Quantitatively evaluating the accuracy of predicted latent hypergraphs in real-world dataset is a very important but also very challenging problem, that remains open in the community. Our datasets lack annotations for higher-order structures, and even if they would have existed, there’s no guarantee they would be optimal for the task. This aligns with challenges in the explainability community, where models are typically tested on synthetic data. Additionally, research in the rewiring community suggests that even when a ground-truth (hyper)graph is available, it is often suboptimal for solving the task. All of these motivated us to construct the synthetic dataset, where we can control a strong connection between the hypergraph structure and the label.
>
> However, to better interpret the type of hypergraph predicted by our model in the real-world datasets, we analyze the NBA dataset, an inductive dataset that is easier to visualize and interpret. For a model with six hyperedges and four nodes per hyperedge:
> - the avg node degree for our predicted hypergraphs are [1.97, 1.98, 1.99, 1.99, 1.98, 1.95, 1.96, 1.96, 1.96, 1.95, 4.29].
> The first 10 nodes correspond to players - model focuses uniformly on all of them; while the last node represents the ball - the model learns to include the ball in most hyperedges. Thus, the predicted structures are star-like hypergraphs, aligning with our intuition that player movement is highly influenced by the ball’s position.
> - the average percentage of duplicated hyperedges across examples is 2.74%. This shows a high diversity in our predictions, confirming that the model captures not just the most likely connections but also learns a dynamic, scene-specific structure.
>
> Regarding the diversity of the datasets, we especially picked them to span both the inductive and transductive setup, with a wide variety of sizes: NBA (11 nodes) ModelNet40 (12311 nodes), NTU (2012 nodes).
>
> ***Model is easy to optimize* without analysis**
>
> As mentioned in the introduction, by *easy to optimise* we mean that our model does not require additional regularisation losses that are necessary in most of the previous works in order to stabilise the training (e.g. sparsity regularisation, reconstruction regularisation). We are sorry for not being clear enough in that respect and we are happy to further clarify this claim if the reviewer thinks it necessary.
>
> **Missing review of classical methods and dynamic graph structures**
>
> We aimed to provide a broad overview, covering 2018-2024 papers from both machine learning and closed-form solution methods. We're happy to expand it and would appreciate any specific suggestions from the reviewer.
>
> In the section *Structural Inference on Graphs*, we discuss advancements in the graph prediction literature. However, we apologize if any relevant works were overlooked and we will do our best to do a more comprehensive review in the final version.
>
> **Analysis of the model’s interpretability**
>
> The interpretable character of our model is indeed given by the discrete structure of our hypergraph which enables: a) explicit inspection of the latent structure used by the final model, and b) discovery of meaningful correlations in the input data. For a concrete example, please refer to the NBA analysis above, which illustrates how our model captures interpretable patterns in real-world data.
>
> **Scalability for large-scale data**
>
> For a model with N nodes and M hyperedges, each containing K nodes, the computational complexity is O(MxN+MxlogNxlogK) . This can become a challenge when both the number of nodes and hyperedges grow significantly. The primary computational bottleneck lies in the slot-attention algorithm. To enhance scalability for large datasets, a potential optimization is to precompute an initial clustering of nodes and limit slot-attention computations to selecting hyperedges within each cluster rather than across the entire dataset. This adjustment would reduce the complexity to O(MxN'+MxlogN'xlogK), where N' represents the maximum cluster size.
>
> Following your suggestions, we measured time and memory per iteration on NBA (batch size 128, 11 nodes) and ModelNet (12,311 nodes) using a Quadro RTX 8000 (49GB). While ensuring architectural comparability, our code is not optimized for large-scale datasets. Further engineering improvements can further enhance scalability.
>
> |Metric|ModelNet40| | |NBA| | |
> |-|-|-|-|-|-|-|
> ||SPHINX |TDHNN||SPHINX|TDHNN|GroupNET|
> |**Training Time (sec)**|0.32 |0.77||0.018|0.21|0.071|
> |**Inference Time (sec)**|0.26|0.76||0.030 |0.22|0.030|
> |**Memory Usage (MB)**|9156.17|20559.32||187.61|508.00|3261.53|
>
> While our model's memory requirements increase with dataset size, it remains significantly more efficient than previous models in the literature.

---

### Official Review · Reviewer_WzEA · 2025-03-17

**Overall Recommendation:** 3

**Summary:**

The paper introduces SPHINX, a novel model designed to infer latent hypergraph structures in an unsupervised manner solely from task-dependent signals. Recognizing the limitations of traditional graph models in capturing higher-order interactions, SPHINX employs a sequential soft clustering approach combined with constrained k-subset sampling to generate discrete hypergraph structures. These structures are compatible with existing hypergraph neural networks and can be optimized end-to-end without additional regularization losses. Through extensive experiments on four challenging datasets, including both synthetic and real-world data, the authors demonstrate that SPHINX effectively infers meaningful hypergraphs that enhance performance in both transductive and inductive tasks, outperforming existing hypergraph prediction methods.

**Claims And Evidence:**

The paper makes several key claims:

1.SPHINX can accurately infer latent hypergraph structures in an unsupervised manner solely from task-dependent signals.
2.The inferred hypergraphs are interpretable and enhance performance in both transductive and inductive tasks.
3.SPHINX outperforms existing hypergraph prediction methods across multiple datasets.

These claims are supported by comprehensive experimental evidence, including ablation studies that highlight the importance of sequential prediction and k-subset sampling. The model is evaluated on both synthetic and real-world datasets, demonstrating superior performance metrics compared to baseline and state-of-the-art methods. Additionally, the paper provides qualitative visualizations of the inferred hypergraphs, aligning closely with ground-truth structures in synthetic settings, further reinforcing the validity of the claims.

**Essential References Not Discussed:**

The paper appears to comprehensively review the relevant literature in the domains of hypergraph neural networks and structural inference. However, it does not mention recent advancements in dynamic hypergraph learning or specific k-subset sampling techniques that might be pertinent. Including references to the latest works in these areas could provide a more thorough context and strengthen the positioning of SPHINX within the current research landscape.

**Experimental Designs Or Analyses:**

The experimental design is robust and well-structured. The authors conduct extensive ablation studies to isolate the contributions of sequential prediction and k-subset sampling, demonstrating their necessity for optimal performance. The use of both synthetic and real-world datasets allows for comprehensive evaluation of the model's capabilities in controlled and practical scenarios. Additionally, the comparison with a range of baseline and state-of-the-art methods across different tasks and datasets ensures that the performance improvements are consistent and significant. The inclusion of qualitative visualizations further aids in understanding the effectiveness of the inferred hypergraphs.

**Methods And Evaluation Criteria:**

The proposed method, SPHINX, integrates a sequential soft clustering mechanism with constrained k-subset sampling to infer hypergraph structures. Specifically, it utilizes a slot-attention mechanism adapted for sequential prediction to address ambiguity issues inherent in parallel hyperedge prediction. The k-subset sampling ensures that each hyperedge contains exactly k nodes, facilitating stable optimization without the need for additional regularization.

For evaluation, the authors employ both synthetic and real-world datasets. The synthetic Particle Simulation dataset allows for direct assessment of hypergraph inference accuracy by comparing predicted hyperedges with ground truth. Real-world datasets, including the NBA SportVU, ModelNet40, and NTU datasets, are used to evaluate the downstream performance of SPHINX in inductive and transductive tasks. Metrics such as Average Displacement Error (ADE), Final Displacement Error (FDE), and overlap with ground-truth hyperedges are utilized to measure performance.

**Other Comments Or Suggestions:**

The paper is well-written and presents a clear narrative of the problem, proposed solution, and experimental validation. Including more diverse real-world applications beyond trajectory prediction, such as social network analysis or biological interaction networks, could further demonstrate the versatility of SPHINX. Additionally, exploring dynamic hyperedge sizes or adaptive hyperedge counts in future work would address some of the current limitations and enhance the model’s applicability.

**Other Strengths And Weaknesses:**

Strengths:
1.Innovation: SPHINX introduces a novel combination of sequential slot-attention and constrained k-subset sampling, addressing key limitations in current hypergraph inference methods.
2.Comprehensive Evaluation: The extensive experiments on both synthetic and real-world datasets, along with thorough ablation studies, provide strong evidence of the model’s effectiveness.
3.Interpretability: By generating discrete hypergraph structures, SPHINX offers enhanced interpretability, allowing for better understanding and visualization of high-order interactions.
4.Applicability: The model’s compatibility with various hypergraph neural network architectures and its performance in both inductive and transductive settings make it versatile for different applications.


Weaknesses:
1.Fixed Hyperedge Size: The requirement of fixed k-node hyperedges may limit the model’s flexibility in scenarios where the size of high-order interactions varies.
2.Predefined Hyperedge Count: The necessity to set a maximum number of hyperedges (M) beforehand could be restrictive in dynamic environments where the number of interactions is not known a priori.
3.Limited Theoretical Insight: The paper focuses heavily on empirical results, with limited discussion on the theoretical underpinnings of why the proposed method works effectively.

**Questions For Authors:**

1.  Have you considered extending SPHINX to handle hyperedges of varying sizes, and if so, what challenges do you anticipate?
2.  How does SPHINX perform with larger-scale datasets in terms of computational efficiency and memory usage?
3. Can you provide more insights on how sensitive the model's performance is to the chosen maximum number of hyperedges (M)?

**Relation To Broader Scientific Literature:**

The paper situates its contributions within the broader context of graph and hypergraph neural networks, as well as structural inference in machine learning. It builds upon existing works in neural relational inference and hypergraph neural networks, addressing their limitations by enabling unsupervised hypergraph inference that is both inductive and transductive. SPHINX distinguishes itself by introducing sequential slot-attention and constrained k-subset sampling, which are not extensively explored in current literature. By comparing with a wide range of related methods, the paper highlights its novel approach to hypergraph structure prediction and its superior performance, contributing significantly to the field of higher-order relational modeling.

**Theoretical Claims:**

The paper does not present significant theoretical claims or proofs. The focus is primarily on the empirical performance of the SPHINX model in inferring hypergraph structures and enhancing downstream task performance. Therefore, there are no theoretical proofs to verify.

---

> ### Author Rebuttal · Authors · 2025-03-31
>
> Thank you for your thorough review and constructive feedback. We appreciate the time you've taken to review our work, and we would like to address your concerns and questions.
>
> **Fixed hyperedge size**
>
> We want to mention that, while the cardinality k is indeed non-learnable, SPHINX does allow hyperedges of different sizes. The coefficient k in the k-sampling algorithm does not affect the number of learnable parameters so it can differ from one hyperedge to another. For simplicity and to reduce hyperparameter search space, we use a fixed k across all hyperedges. However, if a more diverse hypergraph structure is needed, an array of k-values can be provided as input
>
> We agree that a dynamic k could enhance flexibility and is a promising direction for improvement. One possible adaptation is selecting k for each hyperedge based on probability distribution statistics. For instance, defining k as the number of nodes above a probability threshold or the rank at which a certain gap appears in the distribution. While this approach remains non-differentiable w.r.t k, it eliminates the need for pre-defined hyperedge cardinality by allowing dynamic values.
>
> While these adaptations are straightforward, they were not included in the current study. Exploring fully learnable hyperedge cardinality is an interesting future direction..
>
> **Predefined Hyperedge Count / Sensitivity w.r.t. M**
>
> Having a fixed maximum number of hyperedges is indeed a limitation that we are sharing with most of the hypergraph predictor methods in the literature.
>
> The experiments presented in the Appendix (Fig 5)  are designed to understand to what extent this affects the performance of the model on the synthetic setup (where we know what is the real number of hyperedges needed). The results show that, as expected, having a too small M causes a drop in performance while having an M that is larger than the golden standard does not affect the performance.
>
> To better understand this behaviour, we measured, for each hypergraph, the average number of distinct hyperedges predicted when providing more slots than required.  The results are as follow:
>
> | Slots | Unique Hedges | MSE |
> |------------|------------------|-----------|
> | **1 slot**  | 1.00 | 0.00007 |
> | **2 slots** | 1.96 | 0.00002 |
> | **3 slots** | 2.15 | 0.000017|
> | **4 slots** | 2.14 | 0.000007|
>
> We believe that this is a very interesting result, as it shows that when equipped with more slots than necessary the model learns some form of redundancy (by predicting the same hyperedge multiple times) instead of hallucinating fake relations.
>
> **Computational efficiency and memory usage on large-scale data**
>
> For a model with N nodes, M hyperedges, and cardinality of hyperedges K, the complexity of our model is O(M × N + M × logN × logK). This becomes problematic when the number of nodes and the number of hyperedges is simultaneously very large.
> The most computationally intensive component is the slot-attention algorithm. To improve scalability for large-scale datasets, one potential optimization is to precompute an initial node clustering and restrict slot-attention computations to selecting hyperedges within each cluster (local structure) rather than across the entire graph. This would reduce complexity to O(M×N′+M×log⁡N′×log⁡K), where N’ is the maximum cardinality of a cluster.
>
> Following reviewer suggestions, we also measured the time and memory consumption per iteration on NBA dataset (batch size: 128, 11 nodes per example) and the ModelNet dataset (hypergraph with 12 311 nodes). All experiments were conducted on 1 GPU,  Quadro RTX 8000 with 49GB. While we ensured architectural comparability between models, we note that our model has not yet been optimized for large-scale datasets. Further engineering improvements can further enhance scalability.
>
> | Metric | ModelNet40  | | |NBA | | |
> |---------------------------|----------------------|-|-------|-------------------|-------------------|-----------------|
> || SPHINX |TDHNN||SPHINX|TDHNN|GroupNET|
> | **Training Time (sec)** |0.32 |0.77||0.018|0.21|0.071|
> | **Inference Time (sec)** |0.26|0.76||0.030 |0.22|0.030|
> | **Memory Usage (MB)**  |9156.17|20559.32||187.61|508.00|3261.53|
>
> While our model's memory requirements increase with dataset size, it remains significantly more efficient than previous models in the literature.
>
> **Additional  related work on dynamic hypergraphs and k-subset sampling algorithms**
>
> We thank the reviewer for the suggestions. We will do our best to incorporate a review of the k-subset sampling advancement together with  more dynamic hypergraph learning methods. We are welcoming any particular suggestions of relevant work that we are missing.
>
> We appreciate the suggestions and agree that enabling hypergraph inference unlocks numerous real-world applications. While our experiments focused on a subset of topics, we are eager to explore broader real-world applications in future work.

---

### Decision · Program_Chairs · 2025-05-01

**Decision:**

Accept (poster)

**Comment:**

This paper proposes a novel model, i.e., SPHINX, designed to infer latent hypergraph structures in an unsupervised manner solely from task-dependent signals. All four reviewers are positive about this paper with the overall ratings of two accept and two weak accept. The reviewers are generally satisfied with the rebuttal. The AC agrees with the reviewers and recommends acceptance. Please take the reviewers' detailed comments and the rebuttal into consideration when preparing the final version.